# CeGAL: Redefining a Widespread Fungal-Specific Transcription Factor Family Using an In Silico Error-Tracking Approach

**DOI:** 10.3390/jof9040424

**Published:** 2023-03-29

**Authors:** Claudine Mayer, Arthur Vogt, Tuba Uslu, Nicolas Scalzitti, Kirsley Chennen, Olivier Poch, Julie D. Thompson

**Affiliations:** 1Complex Systems and Translational Bioinformatics (CSTB), ICube Laboratory, UMR7357, University of Strasbourg, 1 rue Eugène Boeckel, 67000 Strasbourg, France; 2Faculté des Sciences, Université Paris Cité, UFR Sciences du Vivant, 75013 Paris, France

**Keywords:** fungal-specific transcription factors, genome annotation errors, large-scale biocomputing analysis

## Abstract

In fungi, the most abundant transcription factor (TF) class contains a fungal-specific ‘GAL4-like’ Zn2C6 DNA binding domain (DBD), while the second class contains another fungal-specific domain, known as ‘fungal_trans’ or middle homology domain (MHD), whose function remains largely uncharacterized. Remarkably, almost a third of MHD-containing TFs in public sequence databases apparently lack DNA binding activity, since they are not predicted to contain a DBD. Here, we reassess the domain organization of these ‘MHD-only’ proteins using an in silico error-tracking approach. In a large-scale analysis of ~17,000 MHD-only TF sequences present in all fungal phyla except Microsporidia and Cryptomycota, we show that the vast majority (>90%) result from genome annotation errors and we are able to predict a new DBD sequence for 14,261 of them. Most of these sequences correspond to a Zn2C6 domain (82%), with a small proportion of C2H2 domains (4%) found only in Dikarya. Our results contradict previous findings that the MHD-only TF are widespread in fungi. In contrast, we show that they are exceptional cases, and that the fungal-specific Zn2C6–MHD domain pair represents the canonical domain signature defining the most predominant fungal TF family. We call this family CeGAL, after the highly characterized members: Cep3, whose 3D structure is determined, and GAL4, a eukaryotic TF archetype. We believe that this will not only improve the annotation and classification of the Zn2C6 TF but will also provide critical guidance for future fungal gene regulatory network analyses.

## 1. Introduction

Transcription factors (TF) are essential for the regulation of expression pathways in eukaryotes by binding genomic DNA via a DNA binding domain (DBD), for example, composed of a zinc finger structural motif [1]. The ‘classical’ zinc finger domain coordinates a single zinc atom with a combination of four amino acids, usually cysteine or histidine. However, the Zn2C6 domain (also called Zn(II)2Cys6, Zn2/Cys6 or Zn(2)-Cys(6) binuclear cluster domain) is an atypical zinc finger, where the well-conserved CX{2}CX{6}CX{5,16}CX{2}CX{6,8}C motif contains six conserved cysteines that coordinate two zinc atoms to establish the correct folding of the zinc c1uster domain [2,3]. The Zn2C6 domain defines the GAL4-like Zn2C6-TF family, which is quasi-specific to fungi and observed ubiquitously in all fungal species, where it represents the most abundant TF family in each species [4,5,6,7,8]. Zn2C6-TF is involved in a wide range of functions from primary and secondary metabolisms to multidrug resistance and virulence [4,5,6,7,8,9]. In addition to the DBD, which is generally localized in the N-terminal part, Zn2C6-TF contains a region for activation of the transcriptional machinery. This region, sometimes called the TAD (transactivation domain), is present in many eukaryotic TFs from yeast to humans [10] and is generally found in the C-terminal part of the proteins.

Comparative sequence analyses of the Zn2C6-TF family, initiated in the 1990s, revealed the existence of conserved regions between the Zn2C6 DBD and the TAD [11]. One of these regions, named the MHR (middle homology region) [2], is composed of three conserved motifs involving about 80 amino acids. The MHR (also known as Fungal_trans) was extended to eight consecutive conserved motifs embedded in a large functional domain ranging from 225 to 405 residues [12], which is, as with the Zn2C6 DBD, specific to fungal species and represents the second largest fungal-specific TF class [7]. A mean secondary structure prediction performed on the eight motifs suggested that they are mainly composed of α-helices. Ten years later, the crystal structure of Cep3 [13,14], a yeast kinetochore subunit present in the analysis [12], confirmed that the eight motifs are included in an all-alpha domain, hereafter called the MHD (middle homology domain). To date, Cep3 remains the only experimental 3D structure known for an MHD-containing protein.

The functional role of the MHD remains largely elusive, although it has been postulated that the fungal-specific Zn2C6-MHD TF might correspond to the metazoan nuclear receptors, with the MHD echoing the metazoan ligand binding domain involved in the regulation of the TF activity (notably in an inhibitory function) and/or in the regulation and recognition of effectors [15]. Furthermore, it has been postulated that the MHD might also participate in DNA target discrimination [16,17].

In terms of protein domain organization, the domain pair or bigram [18] composed of the Zn2C6 DBD combined with the MHD is the most frequent in the Zn2C6-TF family. For example, of the 54 Zn2C6-TF from the *Saccharomyces cerevisiae* S288C strain, 44 (81.5%) contain a Zn2C6-MHD domain pair [12]. Strikingly, in the InterPro protein family database [19], approximately one-third of the proteins exhibiting an MHD (InterPro ID: IPR007219) are not predicted to contain a zinc finger motif of the Zn2C6 or C2H2 types (InterPro ID: IPR001138 or IPR013087). These TFs, which apparently lack a DNA binding activity, represent the second largest fungal TF class after the Zn2C6 TF and will be called ‘MHD-only’ hereafter. Except for some rare exceptions [20], MHD-only TFs have not been confirmed experimentally and there is some debate about whether the MHD can act independently. For example, in all experimentally proven TFs listed in the TRANSFAC database (https://genexplain.com/transfac/, accessed on 3 September 2022), the MHD is always located downstream of a DBD [5]. 

A recent genome-wide study of the complement of TF in the fungus *Aspergillus nidulans* revealed numerous discrepancies between the predicted protein sequences and the deduced sequences from experimental transcriptomic data, with approximately 30% of the TFs needing some type of correction [21]. Among the badly predicted TFs, a large majority (78%) concerns the Zn2C6- and/or MHD-containing sequences which frequently exhibit non-predicted or non-processed introns leading to premature stop codons and erroneous sequences. It is interesting that most of the *A. nidulans* MHD-only proteins have a domain with predicted DNA binding (mainly of the Zn2C6 type) after an RNA sequence analysis. These high-throughput experimental results prompted us to reassess the fungal-specific MHD-containing TF family, by developing a domain-centric error-tracking approach that takes into account potential mispredictions of protein sequences.

As a starting point, we collected proteins containing an MHD from three sequence databases with different levels of human expertise involved in the genome annotation process. First, the Saccharomyces Genome Database (SGD) is dedicated to the budding yeast *S. cerevisiae* [22] and provides comprehensive information including protein sequences from a collection of 48 *S. cerevisiae* strain genomes. Second, the UniProtKB/Swiss-Prot database is the expertly curated component of UniProtKB [23]. Third, the UniProtKB/TrEMBL database contains computer-generated annotations for all translations of the EMBL nucleotide sequence entries. We then focused our analysis on the MHD-only TFs by applying a specific error-tracking protocol that uses different DBD–MHD combinations to identify potentially mispredicted genes in available fungal genomic sequences, and especially mispredictions that affected the protein domain organization. 

Our large-scale analysis of almost 17,000 MHD-only TFs allowed us to verify that at least 90% of them possess upstream genomic sequence regions coding for a DBD, mostly of the Zn2C6 type. These results suggest that the vast majority of the MHD-only TF sequences present in public databases result from errors, and that the Zn2C6–MHD domain pair represents a canonical domain signature defining the most predominant family of TFs composed of two fungal-specific domains.

## 2. Materials and Methods

### 2.1. Collection of SGD Sequences

The 44 proteins from the *S. cerevisiae* S288C strain with a Zn2C6–MHD domain organization [12] were identified in the SGD database (Appendix A), and their annotated orthologs in the 47 available strains (Appendix A) were downloaded from the SGD website (http://sgd-archive.yeastgenome.org/sequence/strains/strain_alignments.tar, accessed on 27 April 2022). Genome assemblies for the 47 strains were also downloaded from the SGD website (http://sgd-archive.yeastgenome.org/sequence/strains, accessed on 27 April 2022). Ortholog sequences that did not contain the conserved CX{2}CX{6}CX{5,16}CX{2}CX{6,8}C motif were considered to be potentially erroneous.

For each potentially erroneous sequence, we performed a TBLASTN alignment of the S288C reference protein sequence with the corresponding genome assembly. We then tried to identify the causes of the erroneous sequences. First, if TBLASTN hits were found on multiple scaffolds (with percent identity > 95% and length > 20 amino acids), we assumed that the misprediction was due to a genome assembly issue. If multiple TBLASTN hits (with percent identity > 95% and length > 20 amino acids) were found on a single scaffold, but in different reading frames, we assumed that the misprediction was due to a sequence insertion leading to a frameshift error. If a TBLASTN hit was found with percent identity > 95% and coverage = 100%, we assumed that the misprediction was due to a wrongly predicted start codon.

In order to propose a corrected sequence, a protein sequence was then reconstructed from the TBLASTN hits found on the same scaffold. This corrected protein sequence was searched for the conserved CX{2}CX{6}CX{5,16}CX{2}CX{6,8}C motif.

### 2.2. Collection of UniprotKB Sequences

Fungal proteins were identified in the UniProt 2022_01 database [23] by querying for proteins annotated with the InterPro entry IPR007219: *Transcription_factor_dom_fun* or *Fungal_trans*, which covers the middle homology domain (MHD) specific to these transcription factors. Domain architectures of all proteins containing an MHD were then extracted from the InterPro v86.0 database [19]. The 37,646 UniProt sequences annotated with an MHD, but no DBD, were considered to be potentially erroneous (MHD-only). Potentially erroneous sequences from the reviewed UniProt/Swiss-Prot entries and unreviewed UniProt/TrEmbl entries were processed separately: the 12 UniProt/Swiss-Prot proteins with no DBD were analyzed manually, while the 37,634 UniProt/TrEMBL sequences were input to the error identification protocol (Figure 1) and described in detail in the following sections.

### 2.3. Construction of BLAST Databases of Proteins with Full Domain Architecture

UniProt sequences containing an MHD (IPR007219) in combination with a DBD were used to construct BLAST reference databases. Two BLAST databases were constructed: one for each of the two main DBD types, namely Zn2C6 (IPR001138) and C2H2 (IPR013087), found in this TF family and to which the well-studied GAL4 protein belongs. The Zn2C6 BLAST reference database contained 80,456 sequences, while the C2H2 BLAST reference database contained 6314 sequences.

### 2.4. Extraction of Genomic Sequences

For all potentially erroneous sequences in UniProt, the corresponding genomic DNA sequences were extracted from the Ensembl database [24], when an Ensembl cross-reference was available in the UniProt database. To improve detection of the missing DBD, the full-length gene sequences were retrieved together with an additional 1000 nucleotides upstream of the 5′ end of the gene. For the 37,634 potential error sequences, 16,760 genomic DNA sequences were found in the Ensembl database.

### 2.5. Identification of Nearest-Neighbor Reference Sequences

For each MHD-only sequence, BLASTP searches were performed in the two BLAST reference databases containing Zn2C6 and C2H2 sequences in combination with an MHD. The nearest-neighbor sequences with the required domain combination (i.e., DBD and MHD) were selected if a BLASTP hit was identified with an *E*-value < 0.005. Figure 2 shows the *E*-value distribution of BLASTP hits obtained for all neighbor sequence searches.

### 2.6. Identification of Missing DBD Sequences

For each MHD-only sequence with a BLASTP hit to a nearest-neighbor reference protein, two complementary approaches were implemented to search for the missing DBD sequence. First, a local TBLASTN search was performed in the genomic sequence of the potential error sequence, using the protein DBD sequence segment of the nearest neighbor as a query. TBLASTN alignments with an *E*-value < 0.0001 were taken into account. Second, a global pairwise alignment was performed between the genomic sequence of the MHD-only sequence and the full-length protein sequence of the nearest neighbor, using the ProSplign software developed by the NCBI (https://www.ncbi.nlm.nih.gov/sutils/static/prosplign/prosplign.html, accessed on 12 August 2022). ProSplign is a tool for protein-to-genomic sequence alignment, and it is an integral component of the NCBI Eukaryotic Genome Annotation Pipeline. Genes are first localized on the genomic sequence in a compartmentalization step that starts with computing protein-to-genomic blast hits. These give initial insight into the structure of compartments. Hits are separated into two same-strand sets and then compartments are identified within each strand. To do so, the optimization problem is formally defined in terms of genomic sequence coverage and then solved with a dynamic programming algorithm. ProSplign has been shown to produce accurate spliced alignments and is able to compute alignments of distantly related proteins with low similarity.

Pairwise alignments obtained from TBLASTN and ProSplign were analyzed to identify potential DBD-encoding sequence segments in the erroneous sequences. Finally, the potential DBD-encoding sequence segments were compared to an HMM representing the DBD downloaded from the Pfam protein family database [25]: PF00172 for the Zn2C6 DBD and PF00096 for the C2H2 DBD. To do this, the hhmsearch program from the HMMER suite [26] was used and sequences with an *E*-value < 0.1 were considered as hits. In addition to the hmmsearch *E*-value and to eliminate a number of false positive hits, we also checked for conserved amino acids: for Zn2C6 DBD, two occurrences of the pattern C-x(2)-C (where x is any amino acid) were required, while for C2H2 DBD, one occurrence for each of the C-x(2,4)-C and H-x(3,5)-H patterns was required.

In order to determine whether our sequence curation protocol over-predicted DBDs in the potentially erroneous sequences, we used the same error identification protocol to search for Zn2C6 DBD in the C-terminal region of the proteins (i.e., downstream of the MHD). The results of this analysis are described in the Appendix A.

## 3. Results

### 3.1. MHD-Containing Proteins in the SGD Database

We first analyzed the MHD-containing proteins in the SGD: a database that provides comprehensive integrated information for *S. cerevisiae*. The reference (S288C) strain of *S. cerevisiae* contains 44 proteins with an MHD (Appendix A), with all of them exhibiting a Zn2C6–MHD domain pair [12]. In addition to S288C, the SGD contains genome assemblies and annotations for a further 47 strains of *S. cerevisiae* (Appendix A). For each of these 47 strains, we extracted the annotated orthologs of the 44 S288C Zn2C6–MHD proteins. If all orthologs were conserved in all strains, we would expect a total of 2068 orthologs (44 orthologs from each of the 47 strains), but only 1793 orthologous sequences were found in the SGD (Table 1). In other words, 275 (13%) orthologous sequences were not predicted. Furthermore, for the 1793 predicted sequences, 253 (14%) of them did not contain the conserved CX{2}CX{6}CX{5,16}CX{2}CX{6,8}C motif and were considered to be potentially mispredicted genes.

To investigate the causes of the 253 genes with potential errors, we used the 44 S288C protein sequences to search the corresponding genome assemblies in the SGD using TBLASTN (Table 2).

In the majority of cases, significant hits to genomic regions were found and the protein sequence errors could be linked to genome sequencing or assembly issues. Indeed, for 190 (75%) of the 253 proteins, the S288C protein sequence matched to multiple segments of a single genome scaffold with a sequence identity of at least 95%, although the matching segments were found in different reading frames. These frameshifts were mainly due to the insertion of one or two bases in the genome sequence of the *S. cerevisiae* strain, as compared to the S288C sequence. A further nine sequences were found split over multiple scaffolds. For 46 (18%) of the 253 proteins, the S288C protein sequence matched the genome scaffold with a coverage of 100% and sequence identity of at least 95%, and we concluded that the absence of the Zn2C6 domain was due to a wrongly predicted start codon.

We then tried to correct the 253 erroneous sequences by reconstructing the protein sequence from the TBLASTN genome hits. For 243 (95%) sequences, a complete Zn2C6 domain could be found upstream of the MHD domain (Appendix A). The full-length sequences are provided as a Fasta file. After taking into account the detected gene prediction errors, only ten of the two hundred fifty-three MHD-only proteins remained for which a Zn2C6 domain was not found. These included the nine gene sequences split over multiple scaffolds, which could not be resolved due to the genome assembly issues, and one sequence with a frameshift error (although a manual analysis of this genome sequence indicated a frameshift error affecting one of the conserved cysteines in the Zn2C6 domain).

In order to validate our predictions for the 253 erroneous sequences, we searched for RNA-seq datasets in the NCBI Gene Expression Omnibus (GEO) project corresponding to the different *S. cerevisiae* strains, focusing on the strains with the highest number of potentially mispredicted genes. This transcriptome analysis is described in Appendix A. For each mispredicted gene with a coverage of at least 30 reads, the aligned reads in the region of the gene were manually reviewed, confirming that all the predicted DBD sequences were expressed at similar levels to the MHD portion (Appendix A).

In summary, no convincing evidence of MHD-only proteins was found in any of the 47 *S. cerevisiae* strains analyzed here, and all the identified MHDs located in reliable genome sequence scaffolds were associated with upstream Zn2C6 domains.

### 3.2. MHD-Containing Proteins in the UniProt Database

We then queried the UniProt database for all proteins annotated with the MHD (InterPro ID: IPR007219), resulting in a total of 126,861 proteins, with 126,691 in the unreviewed TrEMBL section and 170 in the reviewed Swiss-Prot section. The MHD-containing proteins had a wide range of domain architectures, with 1905 different architectures listed in the InterPro database, although the most frequent domain pairs were as follows: (i) MHD with a Zn2C6 DBD (IPR001138), (ii) MHD-only, and (iii) MHD with one or two C2H2 DBD (IPR013087), as shown in Figure 3 and Appendix A.

For the 170 proteins from the Swiss-Prot section, nearly 90% contained the Zn2C6-MHD domain pair, although this combination was found in a smaller proportion of the TrEmbl proteins with only 63.4%. Conversely, the proportion of MHD-only proteins lacking an annotated DBD was much higher in TrEmbl (31.6%) than in Swiss-Prot (7.1%).

### 3.3. Manual Analysis of the 12 Swiss-Prot MHD-Only Sequences

Since Swiss-Prot entries are curated by experts, we manually investigated the twelve MHD-only sequences in this database (Appendix A). Where possible, we extracted the corresponding genome sequence from either ENSEMBL [24] or GENBANK [27] databases and translated the genome sequence in the three frames to search for potential DBD encoding regions. This was not possible for four of the twelve sequences. For Q5AR44 and A0A5C1RF03, the gene was located at the start of a contig and the genome region upstream of the annotated gene was not available. For B8NJG5, according to the ENSEMBL database, the upstream gene coded for a small protein coding a Zn2C6 DBD. Finally, A6SSW6 was a shorter protein of length 179 (compared to >500 for the other Swiss-Prot proteins), and the MHD hit in the InterPro database was a partial domain. The genome sequence (FR718884) was annotated as “possibly a relic of a transcription factor”.

For all eight remaining proteins, a potential DBD sequence was found either within the existing annotated gene via alternative splicing, or the proximal 5′ region (<1000 nt) via an alternative start codon or in a new exon (Figure 4).

### 3.4. Automatic Analysis of 16,760 TrEMBL MHD-Only Sequences

Based on the manual analysis of Swiss-Prot described in the previous section, an automatic protocol was developed to analyze potentially erroneous sequences retrieved from the UniProt/TrEMBL database. The first step in the protocol involved identifying the corresponding genomic sequences in the ENSEMBL database. This resulted in a set of 16,760 sequences that were used as input for the main error detection step (see Methods). Two different methods were implemented to locate genomic regions within or upstream of the gene that could encode the missing DBD, using either a local or global alignment approach. Figure 5 shows the number of DBD identified by the two methods.

By integrating the results of the local and global alignment searches, DBD sequences could be proposed for 14,482 (86%) of the 16,760 MHD-only sequences tested (Appendix A). The proposed DBD sequences were distributed in 476 fungal species or strains and are provided as a Fasta file and in the Appendix A with additional information concerning the nearest neighbor used for blast searches, the description, the pathogenicity (against animals or plants), and a complete taxonomic description. Most of these sequences correspond to a Zn2C6 domain (82%), with a smaller proportion of C2H2 domains (4%), which correlates well with the proportions found in the manually curated Swiss-Prot section. To verify the quality of the computer-predicted DBD sequences, we took advantage of a previous study performed on the *Aspergillus flavus* TF proteome by Chang and Ehrlich [28]. By a manual analysis of the genomic region, the authors identified an upstream Zn2C6 domain for 67% of the studied MHD-only TF. Of the 85 DBD sequences we predicted here for *A. flavus* MHD-only TF (Appendix A), 59 sequences (69.4%) were strictly identical to the DBD sequences detected by Chang and Ehrlich, thus highlighting the accuracy of our automatic error-checking protocol.

For the 2278 (13.6%) proteins with no DBD identified by our automatic protocol, we then investigated potential causes for the erroneous sequences. Partial hits, with hmmsearch scores below the defined threshold and part of the conserved Zn2C6 or C2H2 motifs (see Methods), were found in 905 sequences. These might indicate complex exon/intron structures that were partly mispredicted by our protocol (an example is shown in the following section) or might be caused by genome sequencing or assembly issues. For example, undefined regions in the genomic sequences, represented by ‘N’ characters, were found in 1363 of the 2278 proteins. Other reasons for not identifying a DBD include the following: (i) the DBD is located more than 1000 nucleotides upstream of the gene, and (ii) the related sequence is not conserved enough to allow the protein–DNA alignment of the DBD.

### 3.5. Reassessment of Domain Pairs in MHD-Containing Sequences

If the missing DBD sequences proposed here were integrated into the public databases, the number of MHD-only proteins would be reduced from 12 to 4 for Swiss-Prot, and from 16,760 to 2278 for TrEMBL (Figure 6 and Appendix A). More importantly perhaps, this would lead to a significant difference in the distribution of domain pairs present in MHD-containing sequences. In the public databases, this distribution was 65%, 5%, and 30% for Zn2C6–MHD, C2H2–MHD, and MHD-only, respectively. However, our error-tracking protocol indicated that the true distribution was closer to 90%, 6%, and 4% for Zn2C6–MHD, C2H2–MHD, and MHD-only, respectively.

Concerning the phylogenetic distribution of the DBD–MHD domain pair, at the phylum level, MHD domains are present in all phyla except Microsporidia and Cryptomycota (Appendix A). The Zn2C6–MHD domain pair is also present in all phyla except Microsporidia and Cryptomycota, and it is therefore difficult to determine the origin or emergence of this domain pair. In contrast, the C2H2–MHD domain pair is found only in Dikarya (Basidiomycota and Ascomycota).

Interestingly, it has been shown previously that there is a significant difference in the TF repertoire of ascomycete and basidiomycete fungi [8], and in particular that the Zn2C6 family (33%) is much more prevalent than the C2H2 (10%) in ascomycete TF, compared to basidiomycete TF (20% and 15% for Zn2C6 and C2H2, respectively). Despite this overall enrichment of C2H2 in the basidiomycete TF, within the MHD sequences, the proportions of ZN2C6 and C2H2 are similar in both clades (90% and 7% for Ascomycota compared to 92% and 3% for Basidiomycota) (Appendix A).

The level of protein sequence errors is of course dependent on the quality of the genome sequencing, assembly, and annotation. A small number of well-characterized organisms had no MHD-only sequences in the UniProt database, including model organisms such as *S. cerevisiae*, *Yarrowia lipolytica*, or *Ustilago maydis.* Nevertheless, some model organisms had a small number of MHD-only sequences, for example, *Schizosaccharomyces pombe* had twenty-seven proteins annotated with an MHD, of which two proteins had no DBD, or *Candida albicans* with 28 MHD-containing proteins, of which only one had no DBD: Q5AJ63_CANAL (Figure 7).

At the other extreme, *Rhizopus delemar* had thirty-four MHD-containing proteins of which only six were also annotated with a DBD, i.e., 82% were MHD-only proteins. According to our protocol, DBD could be detected for a further twenty-four MHD proteins and only four (12%) lacked a DBD. A further manual analysis of these four MHD-only proteins showed that two genes (I1BR31, I1C782) had regions coding complete Zn2C6 domains within 1500 nt upstream of the 5′ end, while one gene (I1CF81) had a partial hit within the default threshold of 1000 nt upstream of the 5′ end (−490 to −192). According to the public databases, I1CF81 was coded by a gene with one exon and contained one known domain: the MHD. The partial DBD hit for I1CF81 in fact contained a misprediction of a short exon coding for two amino acids, as shown in Figure 8.

## 4. Discussion

In this work, we used a domain-centric in silico approach to show that the second most abundant fungal-specific TF family in the public databases, namely the MHD-only TFs, results largely from genome annotation errors leading to unpredicted DBD. Taking advantage of the high-quality sequences of the *S. cerevisiae* reference strain S288C and the availability of numerous other fungal genomes, we defined an error-tracking strategy involving increasing levels of difficulty: starting with the analysis of the MHD-only sequences present in 48 closely related *S. cerevisiae* strains, followed by the MHD-only sequences in the expert curated Swiss-Prot database, and finally, in the automatically generated TrEMBL database.

The reference *S. cerevisiae* S288C strain has no MHD-only TF coding genes and we showed that, for 95% of the MHD-only TF genes observed in the other *S. cerevisiae* strains, a complete Zn2C6 domain is located upstream of the MHD-coding genomic region. This highlights an unexpectedly high rate of gene prediction errors in such closely related genomes. This high error rate was confirmed for the MHD-only TFs present in the expert curated Swiss-Prot database, since a DBD could be identified for all the proteins whose corresponding genomic sequence was available. Finally, concerning the TrEMBL proteins, our error-tracking protocol showed that 89% of the MHD-only TFs exhibit upstream genomic sequence regions coding for a DBD. These analyses, showing that MHD-only TF sequences result predominantly from prediction inaccuracies, are in line with the error rate of 66% observed in the manual analysis of MHD-only TF in *Aspergillus flavus* [28]. Similarly, a recent high throughput RNA-sequencing experiment in fungal species identified a large proportion of prediction errors in TF sequences [21].

The high rate of wrongly predicted TF sequences (at least 82%) is particularly surprising given that (i) fungal genome sequences are generally of better quality with fewer genome assembly errors, thanks to their relatively small, compact genomes and the low level of repetitive sequences in most fungi [29]; (ii) fungi serve as model eukaryotic organisms and a wide range of diverse genomes have been sequenced and annotated (SGD, Génolevures: genolevures.org, 1000 Fungal Genomes Project: mycocosm.jgi.doe.gov accessed on 27 April 2022); (iii) genome annotation in the fungi is facilitated by the relatively streamlined gene structures and transcriptional processes in these organisms with few and typically short introns rarely implicated in alternative splicing. Our results clearly indicate that all these fungal features, which should promote gene prediction quality, do not limit the error rates at least in the studied TF family. Most importantly, the true number of MHD-only genes remains to be determined, if MHD can indeed act independently [5,7].

The notion of errors in public protein databases is a recurrent problem [30,31,32] and substantial efforts have been invested to identify and correct genome annotation errors [33,34,35]. Some important causes of erroneous protein sequences have been identified, including the genome sequence quality and gene structure complexity [36], as well as redundant or conflicting information in different resources or in the literature [32,37]. Consequently, it has been estimated that 40 to 60% of the protein sequences in public databases are erroneous [38,39,40]. Typical errors include missing exons, non-coding sequence retention in exons, wrong exon and gene boundaries, fragmenting genes, and merging neighboring genes. Our results confirm that genome sequence quality and gene structure complexity are major drawbacks for correct annotation and provide further evidence of the potential of domain-centric approaches to improve automated methods to identify and correct mispredicted protein sequences [39,41,42,43].

It has been established that some domains always co-occur, leading to the concept of associated domains in proteins also called ‘supra-domains’ [44], DASSEM units [45], or domain co-occurrence [46,47].

Our results indicate that the fungal-specific MHD forms part of a synergistic domain pair with a zinc finger DBD, mostly of the fungal-specific Zn2C6 type. As a consequence, the protein sequences exhibiting the ‘supra-domain’ Zn2C6-MHD architecture may define the most widely distributed and abundant fungal TF family, which we propose to name CeGAL after its most characterized members: **Ce**p3, whose 3D structure has been determined, and GAL4, the archetypal fungal TF. This definition will clarify the classification of fungal TFs and will provide better discrimination of the so-called GAL4-like regulators defined according to the presence of a Zn2C6 domain and which include TFs with diverse domain architectures.

Finally, the Zn2C6–MHD combination within the CeGAL family members may have significant consequences for the fungal scientific community. As Zn2C6-TF functions mainly as homodimers or heterodimers, this implies that the number of sequence-specific TFs and the array of control DNA sequences in target genes need to be reconsidered, as well as the degree of combinatorial regulation involved in the wide range of fungal processes controlled by these TFs [4,5,6,7,8]. More importantly, this will also contribute to a better understanding of fungal gene regulatory networks (GRN), that aim to define the complete set of regulatory interactions between TFs and their target genes at a species level. Classically, GRN analyses combine an initial step of the genome-wide characterization of TF families with experimental data related to the transcriptional effects of TF deletion/overexpression, chromatin immunoprecipitation (ChIP)-based TF binding data, protein–protein interactions, or pairs of genes involved in genetic interactions [48]. However, as GRN studies generally exclude proteins lacking a DBD, the complete repertoire of fungal TFs is frequently underestimated. In light of the impressive improvement of the TF specificity prediction tools [49,50,51], we believe that the definition of the CeGAL family combined with the 14,000 DBD sequences provided in this study will permit more robust GRN analyses.

## Figures and Tables

**Figure 1 jof-09-00424-f001:**
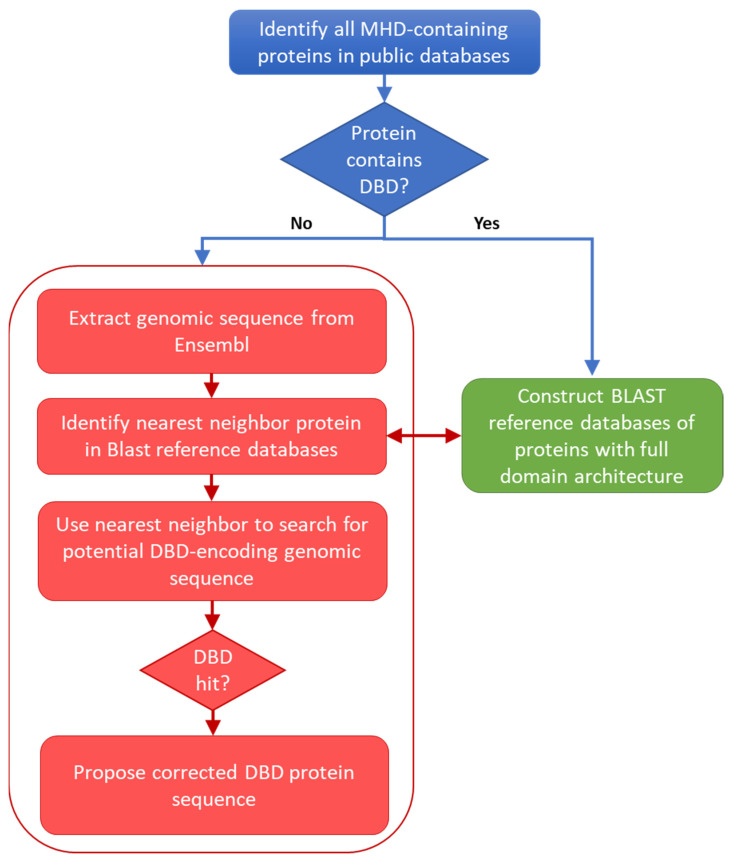
Schema of the protocol used to locate potential errors in sequences retrieved from public databases.

**Figure 2 jof-09-00424-f002:**
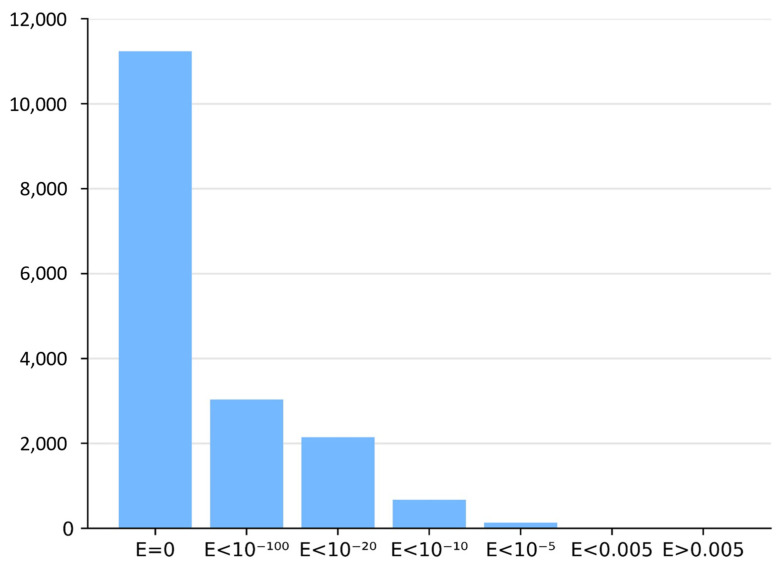
Histogram of BLASTP *E*-values in neighbor identification step.

**Figure 3 jof-09-00424-f003:**
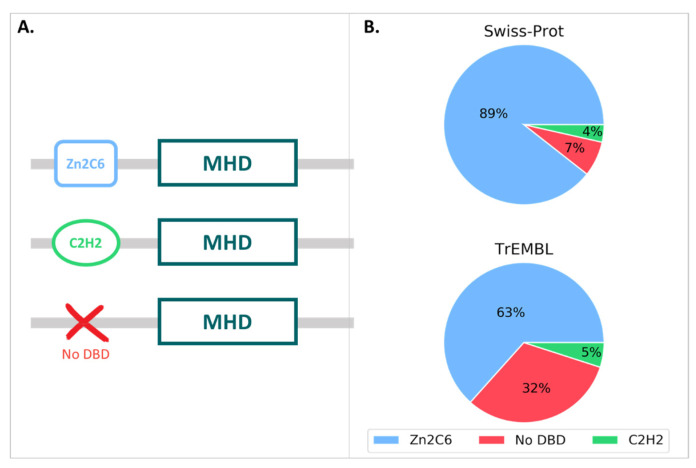
(**A**) MHD domain pairs found in the UniProtKB. (**B**) Proportion of each domain pair found in the Swiss-Prot and TrEMBL sections, with respect to the total number of MHD-containing sequences.

**Figure 4 jof-09-00424-f004:**
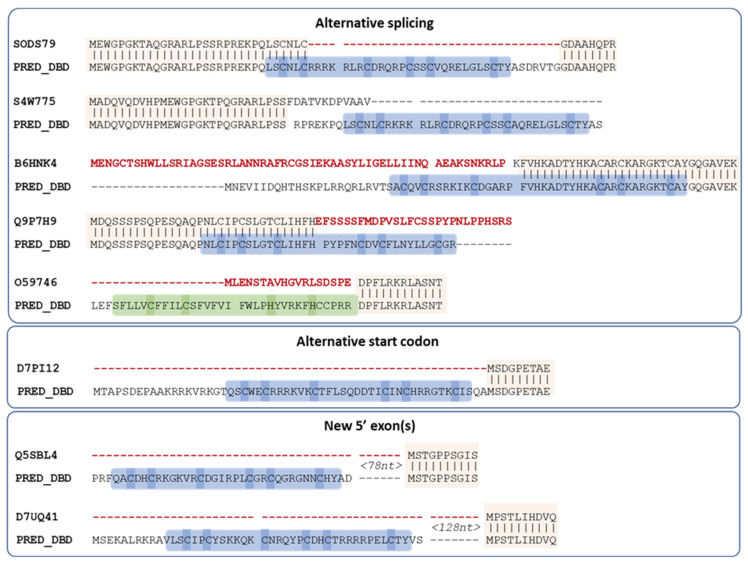
Proposed new sequences for missing DBD of Swiss-Prot MHD proteins. Conserved regions between the existing Swiss-Prot sequence and the proposed sequence are indicated by vertical lines. Regions in the existing Swiss-Prot sequence shown in red are replaced in the predicted sequence, while the predicted DBD is outlined in blue (Zn2C6) or green (C2H2), with cysteines/histidines corresponding to potential zinc binding amino acids highlighted in blue or green. Spaces in the sequences indicate annotated or predicted splice sites.

**Figure 5 jof-09-00424-f005:**
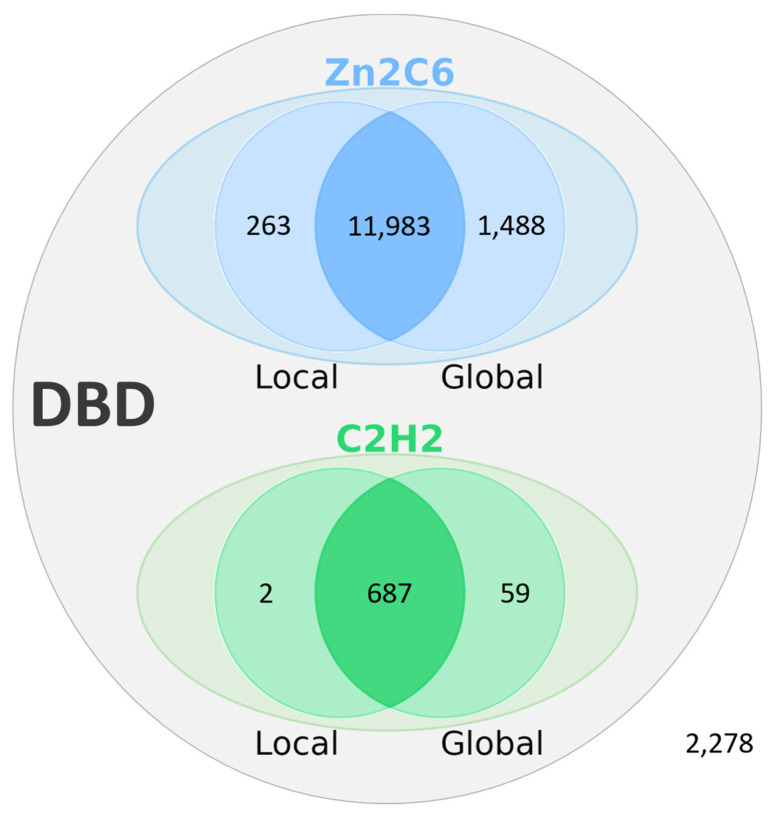
Results of the error identification step in 16,760 MHD-only TF sequences from the TrEMBL database. Number of DBD (Zn2C6 or C2H2) identified by local and global alignment methods. A total of 14,482 MHD-only TF sequences could be attributed to gene prediction errors, while no DBD could be identified for the remaining 2278 sequences.

**Figure 6 jof-09-00424-f006:**
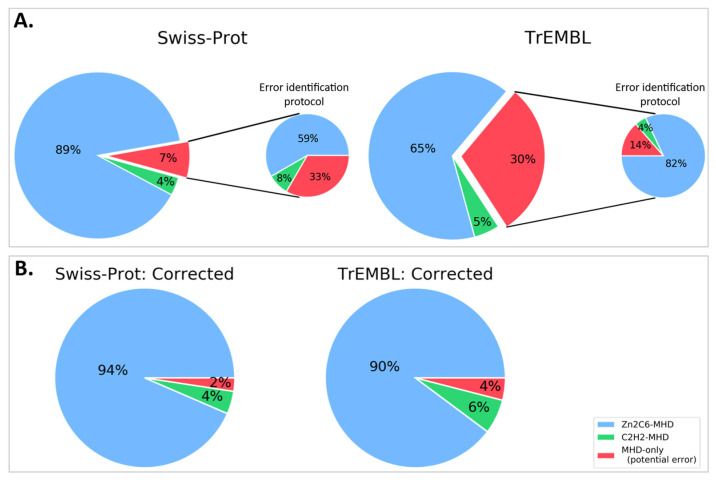
Proportion of sequences with Zn2C6–MHD (blue) or C2H2–MHD (green) domain combinations in (**A**). public databases: Swiss-Prot and TrEMBL (sequences mapped to ENSEMBL only) and (**B**). after applying our error identification protocol. The proportion of potentially erroneous sequences lacking a DBD is shown in red.

**Figure 7 jof-09-00424-f007:**
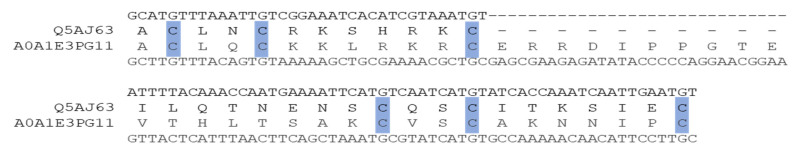
Predicted DBD for *Candida albicans* sequence Q5AJ63_CANAL, aligned with neighbor sequence A0A1E3PG11_9ASCO (hmmsearch *E*-value = 6.5 × 10^−10^). Conserved cysteines characterizing the Zn2C6 DBD are highlighted in blue.

**Figure 8 jof-09-00424-f008:**
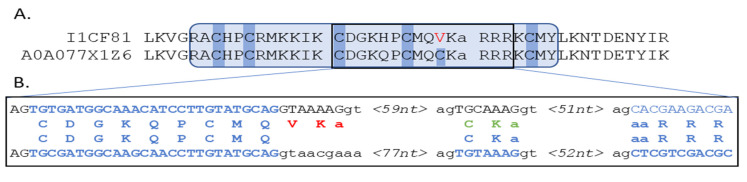
(**A**) Protein alignment of query I1CF81 with the neighbor A0A077X1Z6, showing the partial hit identified by the protocol, where the predicted sequence presents five of the six conserved cysteines that characterize the Zn2C6 DBD (hmmsearch *E*-value = 1.5 × 10^−11^). Exon/intron boundaries are indicated by gaps in the sequences. (**B**) Genome-level comparison of query I1CF81 with the neighbor A0A077X1Z6, showing correctly predicted amino acids (blue), protocol mispredicted amino acids (red), and alternative manual prediction (green).

**Table 1 jof-09-00424-t001:** Domain annotations of MHD-containing proteins in the SGD database.

Domain Annotation	S288C Reference Strain	47 Other Strains
Zn2C6–MHD	44 (100%)	1540 (86%)
MHD-only	0 (0%)	253 (14%)
Total	44	1793

Proportion of sequences with each domain combination, with respect to the total number of sequences (in parentheses).

**Table 2 jof-09-00424-t002:** Probable sources of protein sequence prediction errors in the SGD database.

Error Type	Probable Cause of Error	MHD-Only
Genome sequence error	Frameshift	190 (75%)
	2 or more scaffolds	9 (4%)
Gene prediction error	Wrong start codon	46 (18%)
Undetermined	Undetermined	8 (3%)
Total		253 (100%)

Proportion of each cause of error with respect to the total number of errors detected (in parentheses).

## Data Availability

The genomic and protein sequences supporting the conclusions of this article are available in public databases: SGD, UniprotKB, Ensembl, and GENBANK. The full-length sequences for the corrected MHD-containing proteins from the SGD database and the corrected sequences for the missing DBD in the MHD-containing proteins from the TrEMBL database are provided as supplementary datafiles.

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
