# Peer review of "CeGAL: Redefining a Widespread Fungal-Specific Transcription Factor Family Using an In Silico Error-Tracking Approach"

_jof, 2023, doi:10.3390/jof9040424_

Round 1

Reviewer 1 Report

I have read the manuscript with enthusiasm and joy. I believe that such an analysis was necessary for the fungal community and that it will help in the understanding on how TFs bind target sequences, control gene expression, and how the hierarchical or democratic relationships among TFs in GRNs are established. 

I would only ask if the authors have analyzed and could include in the manuscript information about the evolution of the CeGAL domain or if the emergence of CeGAL and C2H2-MHD TFs could be tracked (when, approximately, the two domains, DBD-MHD, were merged).

Regarding the text, I only have some minor comments, which can be seen in the pdf copy of the document I attach to this report. I did not find the supplementary material together with the manuscript, so I haven´t been able to check it.

Overall, my recommendation is accept with minor review

Author Response

I have read the manuscript with enthusiasm and joy. I believe that such an analysis was necessary for the fungal community and that it will help in the understanding on how TFs bind target sequences, control gene expression, and how the hierarchical or democratic relationships among TFs in GRNs are established. 

I would only ask if the authors have analyzed and could include in the manuscript information about the evolution of the CeGAL domain or if the emergence of CeGAL and C2H2-MHD TFs could be tracked (when, approximately, the two domains, DBD-MHD, were merged).

Response: We have now included an analysis of the phylogenetic distribution of the DBD-MHD domain pair on page 10 and figure S1. At the phylum level, MHD domains are present in all phyla except Microsporidia and Cryptomycota. The Zn2C6-MHD domain pair is also present in all phyla except Microsporidia and Cryptomycota, and it is therefore difficult to determine the emergence of the CeGAL family. In contrast, the C2H2-MHD domain pair is found only in Dikarya (Basidiomycota and Ascomycota).

Regarding the text, I only have some minor comments, which can be seen in the pdf copy of the document I attach to this report. I did not find the supplementary material together with the manuscript, so I haven´t been able to check it.

Response: The comments marked in the PDF have been addressed.

Page 8, line 284. “highlighted in blue” has been changed to “highlighted in blue or green”.

Figure 5. An explanation of the number 2278 has been added to the legend: “A total of 14482 MHD-only TF sequences could be attributed to gene prediction errors, while no DBD could be identified for the remaining 2278 sequences.”

Page 9, line 313. “Of the 85 DBD sequences, we predicted here” has been changed to “Of the 85 DBD sequences we predicted here”.

Page 12, line 448. “this TF” has been changed to “these TF”.

The supplementary materials have now been uploaded to the journal website.

Reviewer 2 Report

Your discoveries could be very important to the society, however, additional steps are needed to verify the results. I have two major concerns about the current version of the manuscript.

1) Please provide more details about the Prosplign step in identification of missing DBD sequences, how could such comparisons can avoid misidentification of DBD in non-coding regions?

2) Please at least verify some of your predictions (e.g. the 253 potentially mispredicted genes in S. cerevisiae ) with transcriptomics data to show the missing DBD actually expressed at similar level as the MHD portion of the gene.

Also, for JoF readers, it would be great if you can provide a brief description of taxonomy and classification for the 17000 MHD-only TF and 14621 re-predicted TFs (instead of line 17-21 in the abstract).

Author Response

Your discoveries could be very important to the society, however, additional steps are needed to verify the results. I have two major concerns about the current version of the manuscript.

1) Please provide more details about the Prosplign step in identification of missing DBD sequences, how could such comparisons can avoid misidentification of DBD in non-coding regions?

Response: The following text has been added to the Methods section. “ProSplign is a tool for protein to genomic sequence alignment, and is an integral component of the NCBI Eukaryotic Genome Annotation Pipeline. Genes are first localized on the genomic sequence in a compartmentization step that starts with computing protein-to-genomic blast hits. These give initial insight into the structure of compartments. Hits are separated into two same-strand sets and then compartments are identified within each strand. To do so, the optimization problem is formally defined in terms of genomic sequence coverage and then solved with a dynamic programming algorithm. ProSplign has been shown to produce accurate spliced alignments and is able to compute alignments of distantly related proteins with low similarity.”

To evaluate the probability of misidentification (over-prediction) of DBD, we used our sequence curation protocol to search for Zn2C6 DBD in the C-terminal region of the 16760 potentially erroneous sequences (i.e. downstream of the MHD). According to the Uniprot database, the MHD-DBD architecture is found in only a very small proportion of MHD-containing TF (0.1%) and therefore we do not expect our protocol to identify many missing DBD sequences in this experiment. Indeed, we found a total of 110 DBD (0.6%) in the C-terminal region. A manual analysis of the 110 genes showed that for 84 sequences (75%), the downstream gene coded for a protein annotated with either a Zn2C6 or a MHD. For a further 6 proteins, the downstream gene coded for other domains already observed in combination with a Zn2C6 in the public databases. Of the remaining 20 sequences, 17 also had N-terminal Zn2C6 domain hits and thus correspond to Zn2C6-MHD-Zn2C6 type architectures.

2) Please at least verify some of your predictions (e.g. the 253 potentially mispredicted genes in S. cerevisiae ) with transcriptomics data to show the missing DBD actually expressed at similar level as the MHD portion of the gene.

Response: We extracted the RNA-seq datasets from the NCBI Gene Expression Omnibus (GEO) project for the S. cerevisiae strains used in this work, focusing on the strains with the highest number of potentially mispredicted genes. Reads were then aligned on the corresponding genome and compared to the gene annotations from the Saccharomyces Genome Database (SGD). For each mispredicted gene with a coverage of at least 30 reads, the aligned reads in the region of the gene were manually reviewed with the Integrative Genome Viewer (IGV) browser, confirming that all the predicted DBD sequences were expressed at similar levels as the MHD portion. This analysis is now described in section 3.1 and a supplementary figure S1 has been added showing screenshots of the IGV browser.

Also, for JoF readers, it would be great if you can provide a brief description of taxonomy and classification for the 17000 MHD-only TF and 14621 re-predicted TFs (instead of line 17-21 in the abstract).

Response: We have now included an analysis of the phylogenetic distribution of the DBD-MHD domain pair. The 17000 MHD-only TF are distributed in 476 fungal species or strains and are found in all fungal phyla except Microsporidia and Cryptomycota. Concerning the 14621 re-predicted TFs, the Zn2C6-MHD domain pair was again found in all phyla except Microsporidia and Cryptomycota, while the C2H2-MHD domain pair was found only in Dikarya (Basidiomycota and Ascomycota).

The abstract (lines 17-21) has been modified: In a large-scale analysis of ~17000 MHD-only TF sequences present in all fungal phyla except Microsporidia and Cryptomycota, we show that the vast majority (>90%) result from genome annotation errors and we were able to predict a new DBD sequence for 14261 of them. Most of these sequences correspond to a Zn2C6 domain (82%), with a small proportion of C2H2 domains (4%) found only in Dikarya. Our results contradict previous findings that the MHD-only TF are widespread in fungi.

Round 2

Reviewer 2 Report

Thanks for your effort in addressing my comments, and I am satisfied with your responses and the revision. This is an important piece of work.